# Integrated Analysis of Transcriptome and Metabolome Provides Insight into *Camellia oleifera* Oil Alleviating Fat Accumulation in High-Fat *Caenorhabditis elegans*

**DOI:** 10.3390/ijms241411615

**Published:** 2023-07-19

**Authors:** Li Liu, Qingbo Kong, Zhuoya Xiang, Xuekun Kuang, Heng Wang, Lijun Zhou, Shiling Feng, Tao Chen, Chunbang Ding

**Affiliations:** 1College of Life Science, Sichuan Agricultural University, Ya’an 625014, China; liuligq@163.com (L.L.); kqb19960308@163.com (Q.K.); xiangzhuoya2015@163.com (Z.X.); 2021216003@stu.sicau.edu.cn (X.K.); 18608352553@163.com (H.W.); zhouzhou124@126.com (L.Z.); fengshilin@outlook.com (S.F.); chentao293@163.com (T.C.); 2Institute of Agro-Products Processing Science and Technology, Sichuan Academy of Agricultural Sciences, Chengdu 610066, China

**Keywords:** camellia oil, *Caenorhabditis elegans*, disorder of lipid metabolism, transcriptome, metabolome

## Abstract

Camellia oil (CO) is a high medicinal and nutritional value edible oil. However, its ability to alleviate fat accumulation in high-fat *Caenorhabditis elegans* has not been well elucidated. Therefore, this study aimed to investigate the effect of CO on fat accumulation in high-fat *C. elegans* via transcriptome and metabolome analysis. The results showed that CO significantly reduced fat accumulation in high-fat *C. elegans* by 10.34% (Oil Red O method) and 11.54% (TG content method), respectively. Furthermore, CO primarily altered the transcription levels of genes involved in longevity regulating pathway. Specifically, CO decreased lipid storage in high-fat *C. elegans* by inhibiting fat synthesis. In addition, CO supplementation modulated the abundance of metabolic biomarkers related to pyrimidine metabolism and riboflavin metabolism. The integrated transcriptome and metabolome analyses indicated that CO supplementation could alleviate fat accumulation in high-fat *C. elegans* by regulating retinol metabolism, drug metabolism—cytochrome P450, metabolism of xenobiotics by cytochrome P450, ascorbate and aldarate metabolism, and pentose and glucuronate interconversions. Overall, these findings highlight the potential health benefits of CO that could potentially be used as a functional edible oil.

## 1. Introduction

Lipids are organic compounds with versatile physiological functions in the body, including structural components, energy storage, signaling, biomarkers, energy metabolism, and hormones [1]. Lipid metabolism disorder is characterized by abnormal levels and quality of lipids and their metabolites in blood, tissues, and organs, caused by congenital or acquired factors. This disorder is associated with numerous metabolic diseases, such as hyperlipoproteinemia, obesity, ketoacidosis, and fatty liver [2]. Currently, lipid metabolism disorders are increasingly prevalent and among the most frequent pathological disorders affecting humans. Despite the availability of drugs for the treatment of lipid disorders, such as statins, fibrates, absorption inhibitors, and nicotinic acid, the prolonged consumption of synthetic drugs may cause severe harm to the liver and kidneys and result in several side effects including nausea, hypoglycemia, diarrhea, constipation, vomiting, headache, decreased appetite, dyspepsia, fatigue, dizziness, and abdominal pain [3]. Therefore, the adoption of lifestyle modifications plays a vital role in mitigating lipid metabolism disorders. The most efficient modification is reducing the intake of saturated fatty acids [4].

Camellia oil (CO) is extracted from the seeds of *C. oleifera* (*Camellia oleifera* Abel.) [5]. It is a highly nutritious and medicinal oil that contains a variety of beneficial ingredients, such as unsaturated fatty acids (UFA), vitamin E, phytosterols, polyphenols, squalene, and trace elements [6]. The Food and Agriculture Organization of the United Nations (FAO) recommends CO as a healthy edible oil [7,8]. Interestingly, the level of UFA in CO reaches more than 90% and it is mainly oleic acid [9]. Oleic acid is known to have numerous health benefits, such as reducing levels of low-density lipoprotein cholesterol, total cholesterol, and glycemic index in the blood [10]. CO has been applied to treat intestinal disorder, burn injury, stomachache and protect the organism against ulcerogenesis, oxidation, and inflammation [11,12]. Long-term intake of CO has been demonstrated to decrease cholesterol levels, lower blood pressure, reduce blood lipid levels, and delay the onset of atherosclerosis [13].

*Caenorhabditis elegans* (*C. elegans*) is a tiny, free-living nematode that is widely used as a model organism [14]. It is an ideal model for genetic research in eukaryotes due to short lifecycle, transparency, and being a self-fertilizing hermaphrodite [15,16]. Between 60% and 80% of human genes have homologues in *C. elegans*, and 40% of genes linked to human diseases have orthologs in the *C. elegans* genome [17]. Therefore, *C. elegans* offers valuable insights into the conserved pathways and underlying mechanisms relevant to human health and disease. *C. elegans* was also the first metazoan organism with a complete genome sequence [18]. Furthermore, it has established complete cell lineage and molecular identification of key genes in developmental and cell biological processes through forward mutation and reverse genetics [19].

In this study, *C. elegans* was used to investigate the mechanism by which CO ameliorates lipid metabolism disorders. The altered metabolic pathways were visualized through an in-depth analysis that integrated transcriptome and metabolome data. These findings emphasize the potential of CO to regulate body fat, which could expand the comprehensive application of CO and has significant implications for the development of the *C. oleifera* industry.

## 2. Results

### 2.1. Quality Parameters and Main Components of CO

As shown in Table 1, the acid and peroxide values of CO were 1.513 mg/g and 0.016 g/100 g, respectively. The major fatty acid composition of the CO can be found in Table 1. From the table, it is evident that the fatty acid composition of CO consists of myristoleic acid (C14:1, 0.01%), palmitic acid (C16:0, 8.55%), stearic acid (C18:0, 1.48%), oleic acid (C18:1, 86.32%), linoleic acid (C18:2, 3.55%), and eicosenoic acid (C20:1, 0.11%), respectively. The CO had an unsaturated fatty acid (UFA) content of 89.99%. Furthermore, the α-tocopherol content of CO was 266.88 mg/kg oil. It should be noted that the β-tocopherol, γ-tocopherol, and σ-tocopherol were not detected in CO (Table 1). The contents of polyphenols and squalene were 51.34 and 26.38 mg/kg, respectively (Table 1).

### 2.2. Effect of CO on Reducing Fat Accumulation in High-Fat C. elegans

In *C. elegans*, the body fat is primarily stored in the intestinal and skin-like epidermal cells [20]. The ORO and TG assays were used to comprehensively evaluate the effect of CO on decreasing fat accumulation in vivo. As shown in Figure 1A, the quantification of staining intensity showed that the ORO staining intensity of glucose-treated worms (Glc, 140.08 ± 5.88) was significantly higher than the Control group (122.86 ± 5.41), confirming the successful construction of the high-fat *C. elegans*. As expected, in CO-treated worms, the ORO intensity (125.59 ± 6.50) was significantly reduced compared to the Glc group. In addition, the ORO staining intensity was significantly reduced by 10.34% and 8.10% after treatment with CO and Orlistat (a saturated derivative of endogenous lipstatin isolated from *Streptomyces toxytricini*, and acts by reversibly inhibiting gastric and pancreatic lipases [21]), respectively. Meanwhile, according to the results obtained by measuring the TG content, the fat accumulation of CO and Orlistat treatment were decreased by 11.54% and 15.16%, respectively. There were some slight differences between TG and ORO methods. Except for TG content, other liposoluble substances were also stained using the ORO method. Under high glucose conditions, the effect of other liposoluble substances will reduce in ORO method because the fat content is increased largely [22].

### 2.3. Effects of CO Supplementation on C. elegans Transcriptome

To further explore the molecular mechanism of CO on reducing fat accumulation in high-fat *C. elegans*, RNA-sequencing analysis was applied to *C. elegans* samples from Control, Glc, and Glc-CO groups. The Pearson correlation coefficients (r) exceeding 92% between different biological repetitions, suggesting extremely strong correlation between samples and highlighting the strong repeatability of the experiment (Figure 2).

Differential expression analysis showed that 86 differentially expressed genes (DEGs) were identified in Glc compared to Control, of which 46 and 40 genes were upregulated and downregulated, respectively (Figure 3A and Appendix A). Relative to Glc, 492 DEGs were detected in Glc-CO (Figure 3B and Appendix A). To identify the possible alleviating fat accumulation genes, the Venn diagram was used to display DEGs in Glc/Control and Glc-CO/Glc. There were 31 common DEGs (Figure 3C).

GO enrichment analysis was carried out to characterize significant functional DEGs. Among these, 86 DEGs were enriched in the 91 GO terms for Glc/Control (*p* < 0.05), consisting of 54 GO terms belonged to biological process (BP), 30 GO terms belonged to molecular function (MF), and 7 GO terms belonged to cellular component (CC) (Appendix A). The protein binding involved in protein folding, protein refolding, misfolded protein binding, chaperone mediated protein folding requiring cofactor, and ‘de novo’ posttranslational protein folding were mainly enriched in Glc/Control (Figure 4A). For Glc-CO/Glc, the DEGs were enriched in the 147 GO terms (*p* < 0.05). Among these significant terms, 85 GO terms belonged to biological process (BP), 42 GO terms belonged to molecular function (MF), and 23 GO terms belonged to cellular component (CC) (Appendix A). The structural constituent of cuticle, defense response, innate immune response, immune response, and immune system process were mainly enriched in Glc-CO/Glc (Figure 4B). It is noteworthy that defense response and collagen trimer were enriched in Glc/Control and Glc-CO/Glc among the top 20 terms of enrichment.

KEGG pathway enrichment analysis of DEGs showed that the biological function 5 and 16 KEGG pathways were significantly enriched in both comparisons (*p* < 0.05, Figure 4C and D, respectively). The 5 enriched pathways of Glc/Control were longevity regulating pathway-multiple species, longevity regulating pathway-worm, endocytosis, maturity onset diabetes of the young, and metabolism of xenobiotics by cytochrome P450. The KEGG top 5 pathways of longevity regulating pathway-worm, longevity regulating pathway-multiple species, drug metabolism-cytochrome P450, metabolism of xenobiotics by cytochrome P450, and spliceosome were enriched in Glc-CO/Glc. In addition, longevity regulating pathway-multiple species and longevity regulating pathway-worm was mainly enriched in two comparisons.

RT-qPCR assays were performed on nine common DEGs to validate the RNA-seq data. The transcriptional patterns detected by RT-qPCR were compared with those obtained from the RNA-sequencing data. As shown in Figure 5, the correlation coefficients (*R*^2^) were 0.6139 and 0.6068, respectively. Indicating that the expression patterns of 9 genes analyzed by RT-qPCR were generally consistent with those observed in the RNA-sequencing analysis. This result confirms the accuracy and reliability of the transcriptome dataset. However, the two methods are slightly different, and may have different sensitivities and detection range.

### 2.4. Effects of CO Supplementation on C. elegans Metabolic Profiles

LC-MS/MS was utilized to explore metabolic variations in *C. elegans* among comparisons of Glc/Control and Glc-CO/Glc. A total of 419 differential expressed metabolites (DEMs) were detected (number of positive and negative ions were 276 and 143, respectively). In positive mode, a total of 179 significant DEMs were identified among Glc/Control with 127 upregulated and 52 downregulated metabolites (Appendix A). For Glc-CO/Glc, 41 metabolites were significantly upregulated and 56 metabolites were downregulated (Appendix A). In negative mode, 110 DEMs were identified among Glc/Control with 59 upregulated and 51 downregulated metabolites. For Glc-CO/Glc, 18 metabolites were significantly upregulated and 15 metabolites were downregulated (Appendix A).

PLS-DA showed significant differences among all groups in both positive and negative models (Figure 6). The corresponding R^2^ values were 0.93 and 0.97 in positive mode and, 0.86 and 0.91 in the negative model, respectively; in addition, Q^2^ was lower than 0 in all comparison groups (Appendix A), indicating the reliability of the proposed models and excellent predictive power.

Combining positive and negative models, a total of 114 annotated metabolites were screened between Glc and Control (Figure 7A). Relative to Glc control, 57 annotated metabolites were screened in Glc-CO (Figure 7B). Simultaneously, 21 metabolites had VIP scores > 2.0 in Glc relative to Control (Figure 7A), such as butylate (VIP = 4.04), montelukast sulfoxide (VIP = 3.10), (z)-n-(1-hydroxy-2-{(z)-[(2e)-1-hydroxy-3-(4-hydroxy-3-methoxyphenyl)-2-propen-1-ylidene]amino}ethylidene) phenylalanine phenylalanine (VIP = 2.89), alonimid (VIP = 2.87), tetracycline (VIP = 2.76), rosinidin (VIP = 2.75), nicotianamine (VIP = 2.57), prostaglandin f1α (VIP = 2.55), ethyl oleate (VIP = 2.43), streptidine 6-phosphate (VIP = 2.43), (2s)-1-(dodecanoyloxy)-3-hydroxy-2-propanyl (9z,12z,15z)-9,12,15-octadecatrienoate (VIP = 2.43), 5′-methylthioadenosine (VIP = 2.40), (3beta,4beta,15alpha,16beta,25s)-4,8,15,16,26-pentahydroxycholest-5-en-3-yl beta-d-xylopyranoside (VIP = 2.39), ophthalmic acid (VIP = 2.28), stepronin (VIP = 2.28), 1-heptadecanoyl-2-palmitoyl-sn-glycero-3-phosphate (VIP = 2.27), N~6~-octanoyllysine (VIP = 2.17), cardiopetalidine (VIP = 2.11), (+)-carpaine (VIP = 2.03), and microcystin lf (VIP = 2.02). Ten metabolites had VIP scores > 2.0 in Glc-CO relative to Glc (Figure 7B), for example, 8222943 (VIP = 2.50), 17 beta-hydroxy-7alpha-mercaptoandrost-4-en-3-one 7-propionate (VIP = 2.34), nobiletin (VIP = 2.31), microcystin-l (VIP = 2.31) and microcystin ly (VIP = 2.19), cytosine (VIP = 2.19), flavin mononucleotide (VIP = 2.06), leukotriene b4 ethanolamide (VIP = 2.05) microcystin la (VIP = 2.02), and 2,3-dinor-11β-prostaglandin f2α (VIP = 2.01).

The KEGG enrichment analysis revealed that the DEMs between Glc relative to Control were mainly enriched in purine metabolism, glycine, serine and threonine metabolism, and valine, leucine, and isoleucine biosynthesis (*p* < 0.05, Figure 8A). For Glc-CO/Glc, there were two KEGG significantly enriched pathways, namely pyrimidine metabolism and riboflavin metabolism (Figure 8B).

### 2.5. Integrated Analysis of Transcriptome and Metabolome

As shown in Figure 9A,B, the DEGs that were highly correlated (|r| > 0.90) with the DEMs in each of the two comparisons. The correlation analysis suggests that these DEGs might play a direct or indirect regulatory role in DEMs metabolism over the high-fat *C. elegans*.

The integrative analysis revealed 2 KEGG enriched pathways in Glc relative to Control, which were glycine, serine, and threonine metabolism and cysteine and methionine metabolism. Therefore, excess glucose exposure in *C. elegans* significantly affected amino acid metabolism.

Additionally, there were 5 KEGG enriched pathways in Glc-CO relative to Glc group. which were retinol metabolism, drug metabolism—cytochrome P450, metabolism of xenobiotics by cytochrome P450, ascorbate and aldarate metabolism, and pentose and glucuronate interconversions. Among them, the retinol metabolism pathway was most significantly enriched. The DEGs that relate to the retinol metabolism pathway including *D2063.1* was significantly upregulated, whereas the *sodh-1*, *ugt-55*, and *ugt-18* were significantly downregulated, while the DEMs of tretinoin were also significantly downregulated (Figure 9E).

## 3. Discussion

The acid value reflects the amount of free fatty acid and the peroxide value indicates the level of oil and fatty acid oxidation [23]. In this study, the acid and peroxide values met the optimal quality standards of the China national standard (GB/T 11765-2018) with an acid value of ≤2.0 mg/g and a peroxide value of ≤0.25 g/100 g [23]. Additionally, CO is rich in unsaturated fatty acids (UFA), tocopherol, polyphenols, and squalene. The abundant UFA in CO is very important in developing and maintaining the nervous system as well as physiological function in body [24]. A-tocopherol has been found to enhance fertility, reduce oxidative stress, alleviate oxidative damage, mitigate neuro-inflammation, and delay aging [25]. Furthermore, polyphenols and squalene are natural antioxidants that scavenge free radicals in the body, effectively preventing tumorigenesis, cardiovascular disease, and lowering cholesterol levels [26,27]. In summary, CO is deemed a high-quality edible oil.

Excessive accumulation of lipids can cause persistent cellular and tissue damage, as well as metabolic diseases, reducing the quality of life [1]. The aim of this study was to investigate the effect of CO (a high-quality edible oil) on glucose-induced lipid accumulation. Supplementation with 0.1 mg/mL CO was found to reduce the accumulation of lipids. CO has shown a dose-dependent capability to decrease high TG, TC, LDL-C, and glucose levels induced by a high-fat diet, while also mitigating inflammation and hepatic steatosis in the liver, decreasing hepatic TG and TC levels [28]. Notably, these outcomes correspond with the effects observed in orlistat treatment.

Transcriptome analysis aimed to investigate the effects of CO on metabolic processes in high-fat *C. elegans*. While glucose serves as the main oxidative fuel source in eukaryotes, excessive glucose availability has detrimental effects on organismal metabolism. Similar to humans, excessively exposed glucose in *C. elegans* results in toxicity that exhibits decreased lifespan, delayed reproduction and fertility, and increased fat accumulation [29,30]. This study found that high glucose conditions increased fat accumulation and deregulated genes involved in the longevity regulating pathway. Enrichment analysis of the KEGG pathway Glc-CO/Glc revealed that CO may alleviate fat accumulation in high-fat *C. elegans* by regulating genes mainly involved in the longevity regulating pathway. Specifically, CO decreased lipid storage in high-fat *C. elegans* by downregulation *fat-7*. The *fat-7* encodes delta-9 desaturases, which function to promote fat stores, and *fat-7* mutants exhibit significantly decreased lipid storage compared to wild type [31,32] Therefore, CO inhibits fat accumulation by downregulating stearoyl-CoA desaturase in *C. elegans*. In addition, CO downregulated the genes in the steroid biosynthesis pathway. Taken together, CO supplementation can alleviate fat accumulation in high-fat *C. elegans* by inhibiting fat synthesis.

Metabolomics is a technique used to detect and identify metabolites in cells and tissues, which is crucial for biological research [33]. In this study, there were 3 KEGG pathways significantly enriched in Glc/Control, namely purine metabolism, glycine, serine, and threonine metabolism, and valine, leucine, and isoleucine biosynthesis, indicating that excess glucose exposure in *C. elegans* significantly affected purine metabolism and amino acid metabolism. Additionally, the supplementation of CO in high-fat *C. elegans* had a significant impact on riboflavin metabolism. Riboflavin (vitamin B2) is a precursor of flavin mononucleotide and flavin adenine, which plays a crucial role in the metabolic reactions and energy homeostasis by regulating mitochondrial pathways including electron transport chain, fatty acid metabolism, and TCA cycle [34]. Riboflavin can enhance antioxidant capacity, inhibit lipid peroxidation, and reduce lipid content [35].

Transcriptomics and metabolomics techniques reflect changes in genotype and phenotype. They provide complementary information about genetic alterations, protein synthesis, metabolisms, and cellular function [36]. Integrative analysis revealed that excess glucose exposure in *C. elegans* significantly affected amino acid metabolism. Amino acids serve as building blocks of proteins and as signals regulating multiple biological pathways, influencing energy balance, and resulting in changes in fat mass and body weight [37]. Therefore, excess glucose helps diminish the amino acid load and increases lipogenesis possibly by affecting the TCA cycle [38,39]. Additionally, integrative analysis demonstrated that CO supplementation alleviate fat accumulation in high-fat *C. elegans* by regulating retinol metabolism. The retinoids (including vitamin A and its biological derivatives such as retinal and retinoic acids or tretinoins) levels are positively correlated with fat storage. In addition, retinoids promote fat unsaturation in response to high-glucose stress and improve organism survival [40].

## 4. Materials and Methods

### 4.1. Materials

The CO was extracted from *C. oleifera* seeds by aqueous extraction method according to previous research in our laboratory. Briefly, the seeds were dried in constant temperature drying oven at 60 °C, and then the dried *C. oleifera* seeds were ground to a fine powder. The ratio of seeds powder to water was 1:4.5 (g/mL). Then, the mixture was shaken at 75 °C (160 rpm) for 150 min in a thermostat water bath. The resulting mixture was centrifuged to separate the oil and stored in the dark at 4 °C for the following assays [41]. All the chemicals used in the study were procured from either Chengdu Kelong Chemistry Co., Ltd. (Chengdu, China) or Nanjing Jiancheng Bioengineering Institute (Nanjing, China).

### 4.2. Quality Parameters and Main Components of CO Assay

The acid value and peroxide value of CO were measured in accordance with China’s national standard (GB5009.229-2016 and GB5009.227-2016, respectively) [42,43].

The fatty acid composition was determined using gas chromatography–mass spectrometry (GC-MS) after transesterification [41]. Briefly, 100 mg of oil was mixed with 2 mL of 1 mol/L NaOH-methanol and shaken for 30 min at 40 °C. The resulting methyl esters were then extracted with 2 ml of n-hexane. The GC-MS analysis was performed using an Agilent 7890A gas chromatograph and a 5977C mass spectrometer (Agilent Technologies, Palo Alto, CA, USA) with a capillary column HP-5MS (30 m × 0.25 mm; 0.25 μm). The oven temperature was programmed from 60 °C (held for 2 min) to 150 °C (held for 2 min) at a rate of 15 °C/min, and then increased to 280 °C at a rate of 15 °C/min for 3 min. Helium was used as the carrier gas with a flow rate of 0.6 mL/min. The injector and detector temperatures were set to 240 °C and 260 °C, respectively. The mass scans were performed in the range of 50 to 500 *m/z*.

The tocopherol content was measured using a previously established method [44]. A sample containing 1 g of oil was dissolved in 10 ml of n-hexane, mixed, and then filtered using a 0.22-mm microporous membrane for high-performance liquid chromatography HPLC (Agilent Technologies, Palo Alto, CA, USA) equipped with a ZORBAX SB-C18 column (150 mm × 4.6 mm, 5.0 μm) analysis. The following detection conditions were applied: a fluorescence detector with excitation wavelength set at 295 nm and emission wavelength at 325 nm, a mobile phase of methanol with a flow rate of 0.8 mL/min, and a column temperature of 35 °C. 

Squalene content was determined based on a modified version of a previously established method [41]. Firstly, the oil was saponified using a potassium hydroxide–ethanol solution, and then the sample was analyzed by HPLC. The following detection conditions were implemented: an ultraviolet detector with a wavelength of 325 nm, a column temperature of 30 °C, and a mobile phase of methanol:acetonitrile (60:40, *v:v*) with a flow rate of 1.0 mL/min.

For the polyphenol extraction, a modified version of a previously described method was used [45]. A 0.5 g sample of oil was combined with 5 mL of n-hexane. To extract the oil, 2 mL of aqueous methanol (80%, *v*/*v*) was added and the extraction process was repeated three times. The extract was then kept at room temperature overnight. The remaining oil was removed using n-hexane. The resulting extract was concentrated using a rotary evaporator and adjusted to a final volume of 10 mL with methanol. To determine the polyphenol content, 0.1 mL of the extract was mixed with 0.02 mL of Folin–Ciocâlteu reagent and 0.08 mL of a 10% sodium carbonate solution for 5 min. Afterwards, 0.8 mL of distilled water was added. The absorbance of the mixture was measured at 765 nm using a Spectra Max M2 microplate reader (Molecular Devices Corp., Silicon Valley, CA, USA) after incubating it in the dark for 1 h. The amount of polyphenols was reported as milligram gallic acid equivalents per kilogram.

### 4.3. C. elegans Strains and Maintenance

The wild-type N2 (Bristol) were obtained from the Caenorhabditis Genetics Center (University of Minnesota, USA). They were grown and maintained in nematode growth media (NGM) with a food source of live *Escherichia coli* OP50 at 20 °C. Synchronized *C. elegans* nematodes were obtained via bleaching solution (5 M NaOH:5% NaClO, *v*:*v*, 2:1). The control group (Control) were cultured on NGM plates containing 0.1% dimethyl sulfoxide (DMSO). The high-fat *C. elegans* were cultured as previously described with some modifications [22]. Briefly, the synchronized nematodes were grown on NGM (containing 1 mM glucose and 0.1% DMSO) with a layer of OP50 (containing 1 mM glucose 0.1% DMSO) as food source at 20 °C (Glc). The CO group (Glc-CO) and Orlistat group were cultured on NGM plates containing 1 mM glucose along with 0.1 mg/mL CO or 6 μg/mL Orlistat (the CO and Orlistat dissolved in DMSO, DMSO final concentration 0.1%). Within this study, all chemicals in NGM plates and liquid medium were expressed as the final concentrations.

### 4.4. Lipid Droplet Visualization by Oil Red O Staining

Oil Red O staining to detect the lipid droplets was performed as previously described [46]. Briefly, synchronization worms aged 60 h were collected and washed thrice with M9 buffer, then fixed with 100 μL 1% paraformaldehyde for 30 min. After fixation, the worms were washed with M9 buffer to remove residual paraformaldehyde. The worms were resuspended and dehydrated in 60% isopropanol for 30 min at room temperature. Subsequently, worms were stained with Oil Red O staining solution (60% Oil Red O stock solution: 40% water) at room temperature for 10 h in the dark. The dyes were removed, and washed with M9 buffer three times. Finally, the stained worms were mounted and imaged using a fluorescence microscope (Nikon DS-Ril, Nikon Corporation, Tokyo, Japan). Image J software (V1.8.0.112) was applied to density quantification analyze. At least 30 nematodes were examined per treatment and carried out in triplicate independently.

### 4.5. Triglycerides (TG) Content Assay

TG test kit (A110-1-1) and total protein quantification kit (A045-4-1) was used to measure the content TG and protein in *C. elegans* according to the manufacturer’s instructions (Nanjing Jiancheng Bioengineering Institute, Nanjing, China). More than 1000 nematodes were examined per treatment and carried out in triplicate independently.

### 4.6. RNA Extraction and Transcriptome Sequencing Analysis

The synchronization worms aged 60 h were collected and washed with M9 buffer, and three biological replicates were used for each sample. RNA extraction and transcriptome sequencing analysis of all samples were performed by Wuhan Huada Gene Technology (BGI, Wuhan China, www.bgi.com, accessed on 1 January 2023). The total RNA from the worms was extracted using a Qubit RNA XR Assay Kit according to the instructions (Thermo Fisher Scientific, Carlsbad, CA, USA). RNA extraction was quantified and assessed for integrity using NanoDrop (Thermo Fisher Scientific, USA) and a 2100 Agilent Bioanalyzer (Agilent, Santa Clara, CA, USA) prior to subsequent experiments. To construct the BGI-based mRNA-seq library, and 1 μg qualified RNA from each sample was used. Then, the library was sequenced on the BGISEQ platform [47]. The raw reads were filtered by SOAPfilter, and an average of 6 Gb high-quality data was generated for each sample. Then, the resulting high-quality reads from each sample were mapped against the *C. elegans* reference genome using HISAT and Bowtie2 tools [48]. The expression level of each gene in each sample was calculated as the FPKM. For each transcription region, FPKM values were calculated by using RESM software (V1.1.17). The differential expression genes (DEGs) were identified using DEseq2 [49]. based on the read’s matrix, with a filter criteria of fold change (FC) > 1.5 and *p* < 0.05. Gene Ontology (GO, http://geneontology.org/, accessed on 1 January 2023) and the Kyoto Encyclopedia of Genes and Genomes (KEGG, https://www.kegg.jp/, 1 January 2023) tools were used to analysis the DEGs. 

### 4.7. Reverse Transcription and Quantitative Real-Time PCR (qRT-PCR) Analysis

RT-PCR was employed to determine mRNA expression of common DEGs in all groups. The synchronization worms aged 60 h were collected and washed with M9 buffer, and then instantly frozen in liquid nitrogen and ground. Total RNA was extracted using a kit (RNA Easy™ Plus, Beyotime, Nantong, China). The quality and quantity of RNA were examined by agarose gel electrophoresis and ultramicrospectrophotometer (NanoDrop 2000, Thermo Fisher Scientific, USA). Reverse transcription and polymerase chain reaction were conducted by prime script RT kit (Takara, China). The targeted gene was measured using SYBR Green (Takara, China) on CFX96 Real-Time PCR instrument (Bio-Rad, USA). Expression was normalized to *act-1* and calculated using 2^−ΔΔCt^. The primers sequences are listed in Appendix A.

### 4.8. Untargeted Metabolomics Analysis

The synchronization worms aged 60 h were collected and washed with M9 buffer, and six biological replicates were used for each sample. The untargeted metabolomics analysis was conducted by Wuhan Huada Gene Technology (BGI, China) following their standard procedures. A pooled quality control (QC) sample was prepared by mixing equal volumes of each extracted supernatant and injected every 10 samples. The metabolites were separated and detected using Waters 2D UPLC (waters, USA) tandem Q exactive high-resolution mass spectrometer (Thermo Fisher Scientific, USA) [50]. Chromatographic separation was conducted using a BEH C18 column (1.7 μm; 2.1 mm × 100 mm, Waters, Milford, MA, USA). In the positive ion mode, the mobile phase was an aqueous solution containing 0.1% formic acid (A) and 100% methanol containing 0.1% formic acid (B), and the negative ion mode mobile phase was an aqueous solution containing 10 mM ammonium formate (A) and 95% methanol containing 10 mM ammonium formate (B). The linear gradient program set as follows: 2% B (0–1 min), 2–98% B (1–9 min), 98% B (9–12 min) 98–2% B (12–12.1 min), 2% B (12.1–15 min). The flow rate was set at 0.35 mL/min and the column temperature was 45 °C. The injection volume was 5 μL. The mass spectra ranged from 70 to 1050 mass-to-charge ratio with primary resolution of 70,000 and secondary resolution of 17,500. The sheath gas flow rate was 40, and the aux gas flow rate was 10. The spray voltages were 3.80 (positive ion) and 3.20 (negative ion). The capillary temperature was 320 °C and the aux gas heater temperature was 350 °C [51]. The raw data were imported into Compound Discoverer 3.1 (Thermo Fisher Scientific, USA) for data processing. The metabolites were identified using a combination of BGI Library (HuaDa’s self-built standard library), mzCloud, and ChemSpider (HMDB, KEGG, LipidMaps). The results exported by Compound Discoverer 3.1 were imported into metaX for data preprocessing. Statistical significance (*p* value) was calculated by univariate analysis (*t*-test). The metabolites with VIP > 1, *p* value < 0.05, and fold change (FC) ≥ 1.2 or ≤0.833 were differentially expressed metabolites (DEMs). Volcano plots were used to filter metabolites of interest based on log2 (FC) and −log10 (*p* value) of metabolites. Based on KEGG database, metabolic pathway enrichment analysis was carried out. Metabolic pathway analysis was performed using MetaboAnalyst 5.0. The pathway with *p* value < 0.05 was considered to be a differential metabolic pathway [52].

### 4.9. Statistical Analysis

Data of ORO intensity and TG content were expressed as means ± standard deviation (SD). Data analysis was performed using SPSS 22.0 version (SPSS, Stanford University USA). Multiple group comparisons were conducted using one-way analysis of variance (ANOVA). Differences were considered significant when *p* < 0.05.

## 5. Conclusions

In this study, CO supplementation alleviated fat accumulation in high-fat *C. elegans*. It mainly altered transcription levels of genes involved in fat synthesis, and modulated the abundance of metabolic biomarkers involved in pyrimidine metabolism and riboflavin metabolism. Integrated analysis of transcriptome and metabolome data indicated that CO supplementation alleviates fat accumulation in high-fat *C. elegans* by regulating retinol metabolism, drug metabolism—cytochrome P450, metabolism of xenobiotics by cytochrome P450, ascorbate and aldarate metabolism, and pentose and glucuronate interconversions. This study enhances the understanding of CO’s ability to reduce lipid accumulation. Therefore, it provided a theoretical basis for the study of the efficacy of CO. Nevertheless, there are some limitations in this work, including that the functions of many DEGs and DEMs are poorly reported in the literature and the small sample size in the metabolomic analysis. In addition, a more detailed profile of CO utilization in vivo is required in future studies.

## Figures and Tables

**Figure 1 ijms-24-11615-f001:**
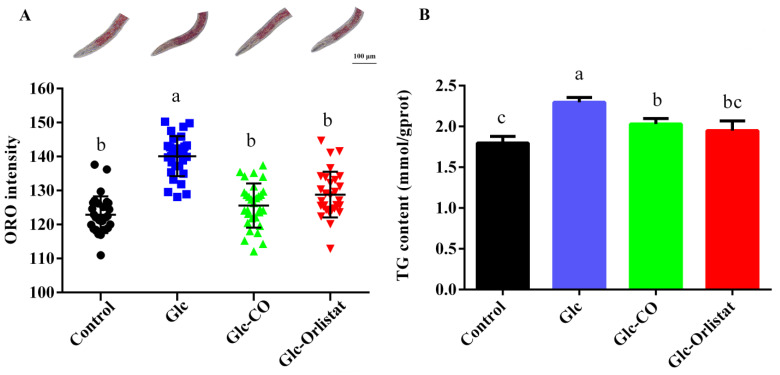
Effect of CO on reducing fat accumulation in high-fat N2. (**A**) ORO staining of *C. elegans*. (**B**) The TG content of *C. elegans*. Letters (a, b, c) indicate significant differences determined by one-way ANOVA test followed by Dunnett’s post-hoc test for selected groups (*p* < 0.05). Sample size: *n* = 30. Results represent mean ± SD.

**Figure 2 ijms-24-11615-f002:**
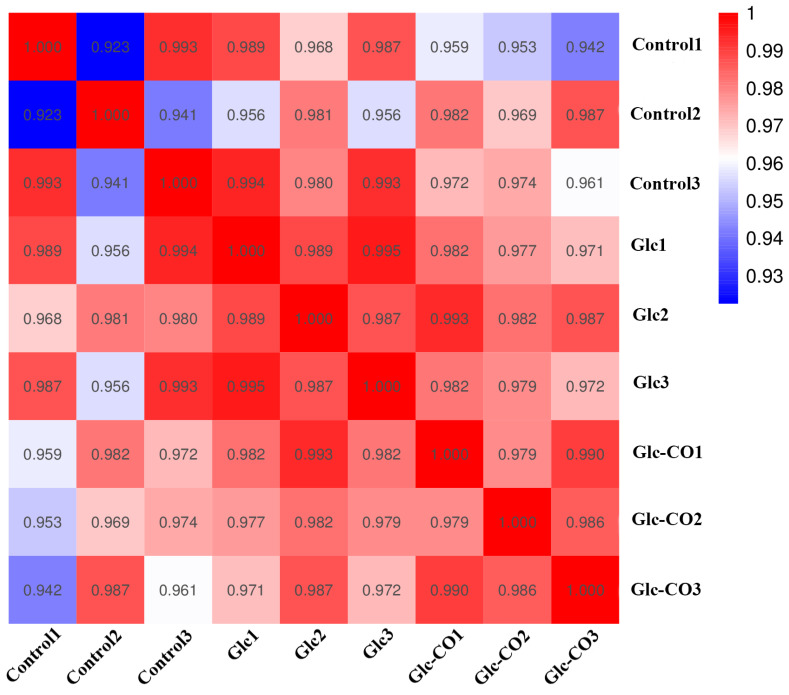
The Pearson correlation coefficients analysis between different biological repetitions.

**Figure 3 ijms-24-11615-f003:**
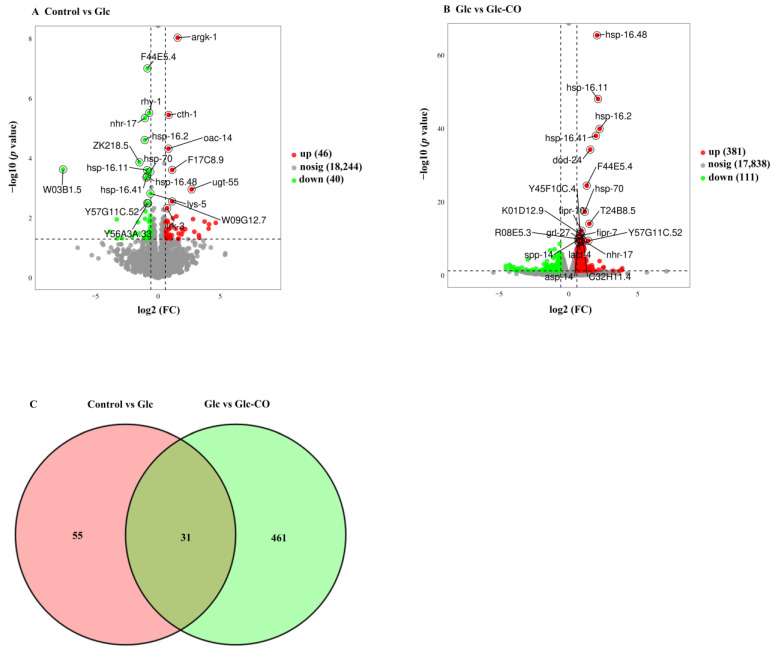
Volcano plot of significant DEGs of (**A**) Glc/Control, (**B**) Glc-CO/Glc, Red dots, upregulated; green dots, downregulated; gray dots, no significance. (**C**) Venn diagram of DEGs in Glc/Control and Glc-CO/Glc.

**Figure 4 ijms-24-11615-f004:**
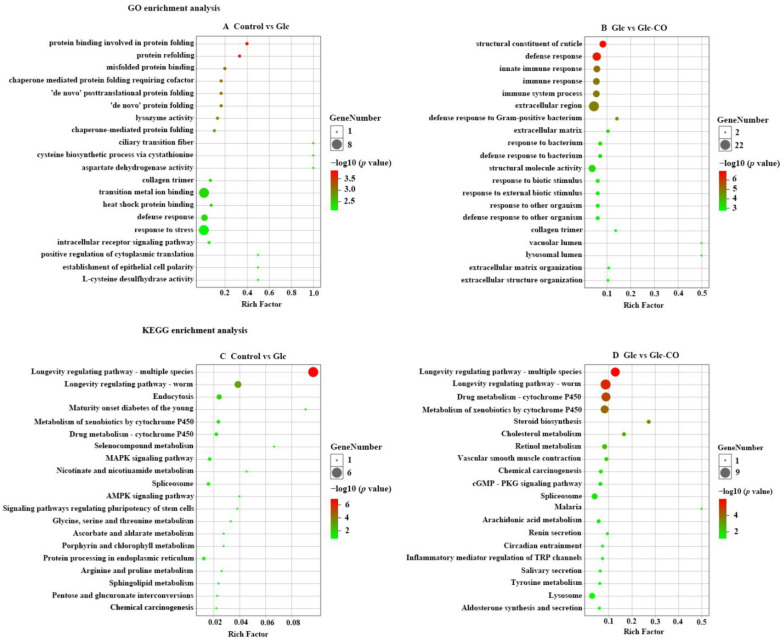
GO enrichment analysis of (**A**) Glc/Control, (**B**) Glc-CO/Glc, bubble diagram size of dots represents genes number in each GO term. KEGG enrichment analysis of (**C**) Glc/Control, (**D**) Glc-CO/Glc. The size of the circles corresponds to the number of DEGs and are color-coded according to −log10 (*p* value). The *x*-axis shows the enrichment factor value.

**Figure 5 ijms-24-11615-f005:**
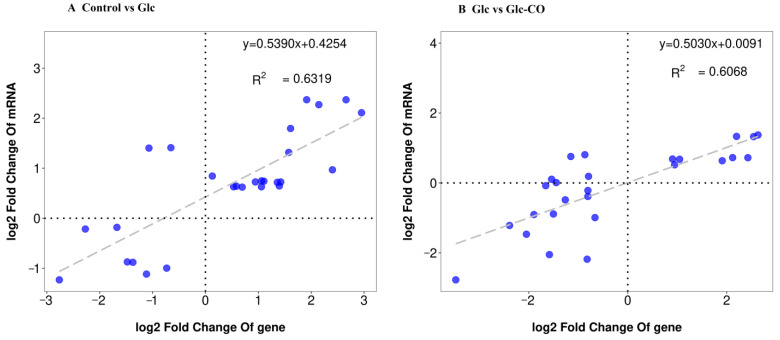
Correlation analysis based on RNA-seq and RT-qPCR data. (**A**) Glc/Control, (**B**) Glc-CO/Glc.

**Figure 6 ijms-24-11615-f006:**
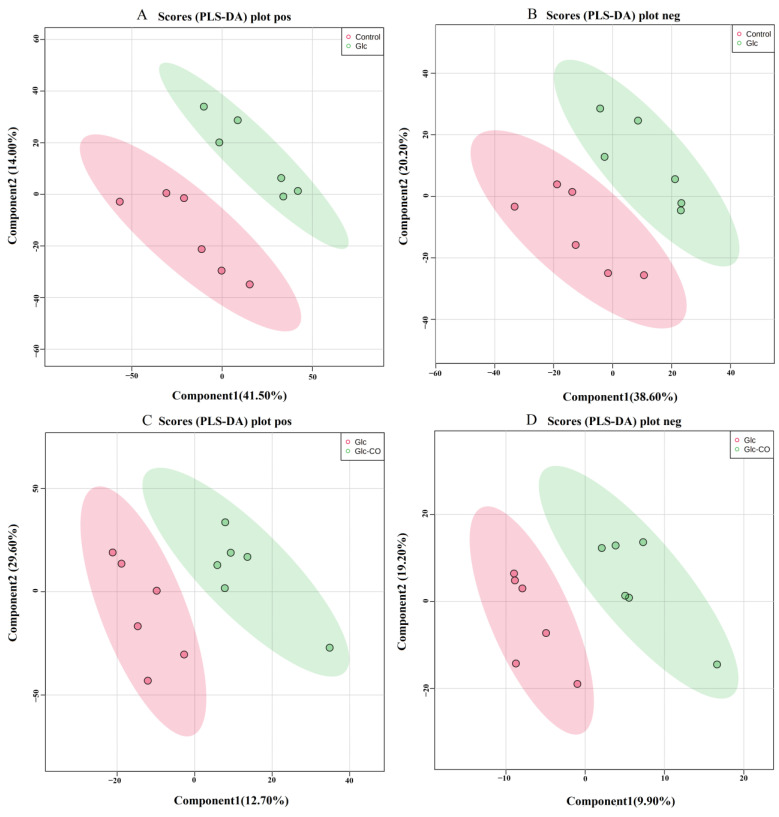
PLS-DA plot of (**A**) Glc/Control pos, (**B**) Glc/Control neg, (**C**) Glc-CO/Glc pos, (**D**) Glc-CO/Glc neg. pos: positive ion, neg: negative ion.

**Figure 7 ijms-24-11615-f007:**
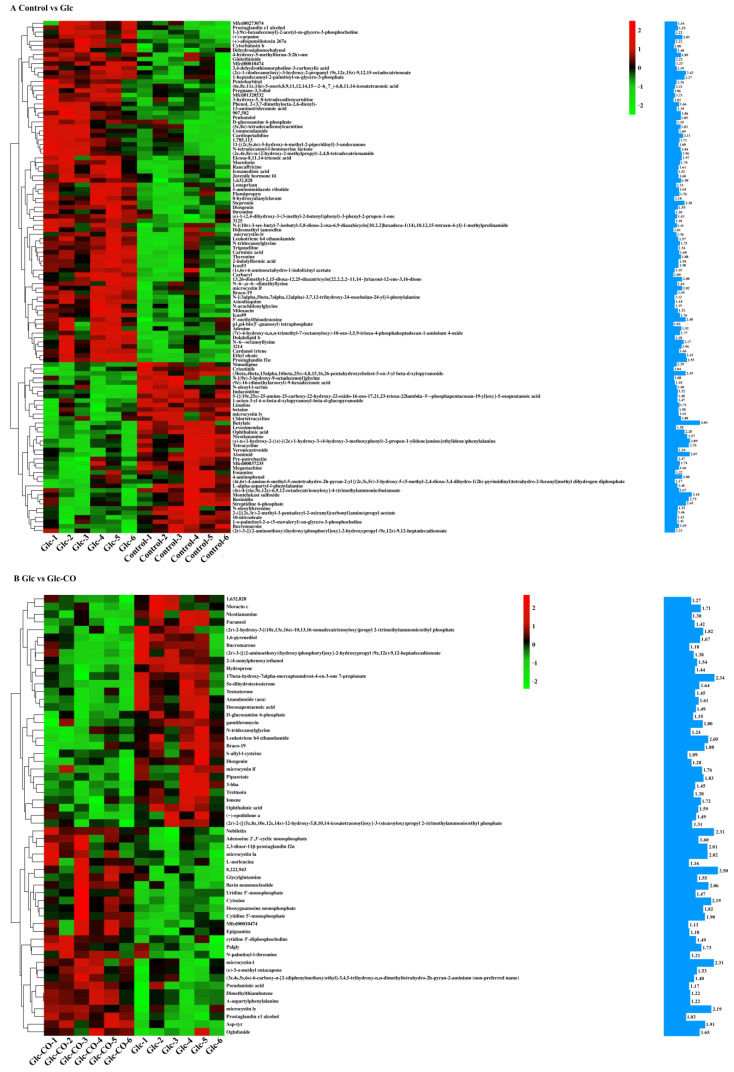
The expression profile and VIP of metabolites. (**A**) Glc/Control, (**B**) Glc-CO/Glc. Color indicates level of relative content of each DEM, from green (low) to red (high). The blue histogram indicates the VIP value.

**Figure 8 ijms-24-11615-f008:**
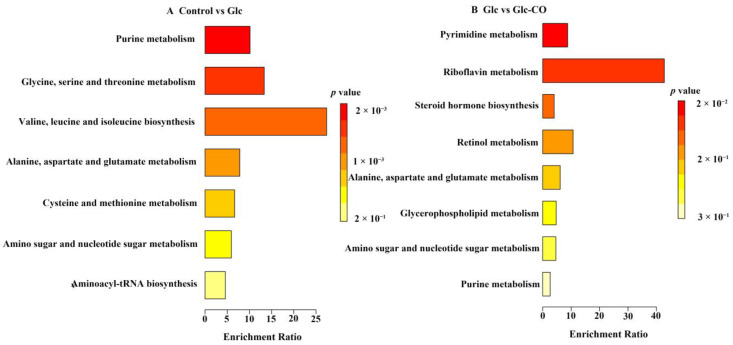
KEGG enrichment analysis of metabolites. (**A**) Glc/Control, (**B**) Glc-CO/Glc. The *x*-axis shows the enrichment factor value and are color-coded according to *p* value.

**Figure 9 ijms-24-11615-f009:**
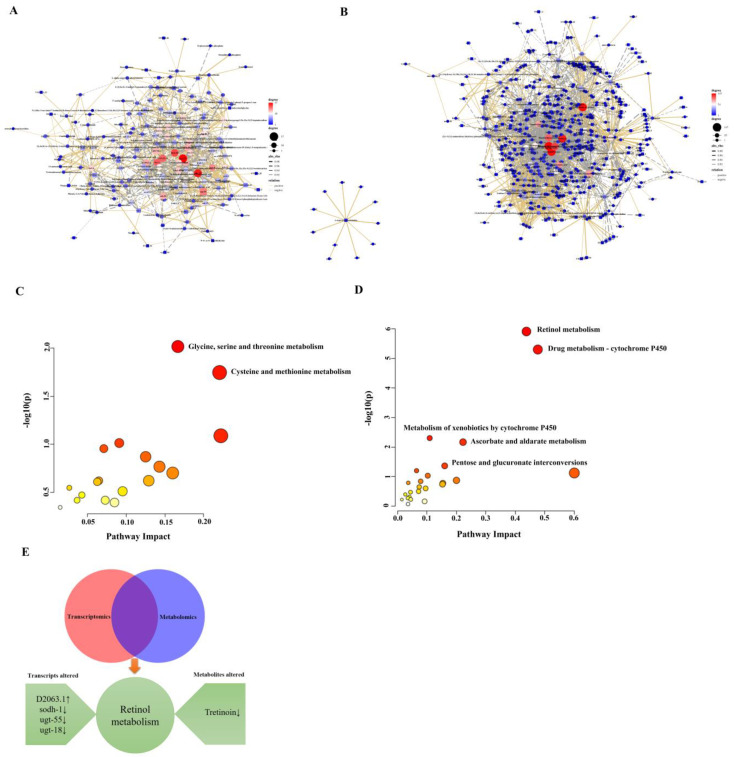
Integrated analysis of metabolomic and transcriptomic data. Correlation analysis of (**A**) Glc/Control, (**B**) Glc-CO/Glc. Edges represent “relationships” between any DEMs and DEGs, yellow and gray represent positive and negative correlations, respectively, as determined by absolute value of |r| > 0.90, DEMs and DEGs of Glc/Control interacted by 1069 edges and 188 nodes; DEMs and DEGs of Glc-CO/Glc interacted by 2860 edges and 497 nodes. Bubble diagram of enriched KEGG pathways of (**C**) Glc/Control, (**D**) Glc-CO/Glc. The *x*-axis shows the enrichment factor value. The size of the circles corresponds to the number of DEGs and DEMs, and are color-coded according to −log10 (*p* value). (**E**) Integrated analysis of transcriptome and metabolome the most significantly enriched pathway of Glc-CO/Glc.

**Table 1 ijms-24-11615-t001:** Quality parameters and main components of CO.

	CO
Acid value (mg/g)	1.513 ± 0.089
Peroxide value (g/100 g)	0.016 ± 0.001
Myristoleic acid (C14:1, %)	0.01 ± 0.01
Palmitic acid (C16:0, %)	8.55 ± 0.34
Stearic acid (C18:0, %)	1.48 ± 0.11
Oleic acid (C18:1, %)	86.32 ± 1.22
Linoleic acid (C18:2, %)	3.55 ± 0.66
Eicosenoic acid (C20:1, %)	0.11 ± 0.04
UFA	89.99
SFA	10.03
UFA/SFA	8.97
α-tocopherol (mg/kg)	266.88 ± 9.91
β-tocopherol (mg/kg)	-
γ-tocopherol (mg/kg)	-
δ-tocopherol (mg/kg)	-
Polyphenols (mg/kg)	51.34 ± 4.19
Squalene (mg/kg)	26.38 ± 0.88

Data are expressed as the mean ± standard deviations except UFA, SFA, and UFA/SFA, *n* = 3. The “-” represents not detected in the identification process. “UFA” represents unsaturated fatty acid. “SFA” represents saturated fatty acid.

## Data Availability

Data are available on request.

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
