# Peer review of "Integrated Analysis of Transcriptome and Metabolome Provides Insight into Camellia oleifera Oil Alleviating Fat Accumulation in High-Fat Caenorhabditis elegans"

_ijms, 2023, doi:10.3390/ijms241411615_

Round 1

Reviewer 1 Report

Review of Liu et al (2023), “Integrated analysis of transcriptome and metabolome provides insight into the Camellia oleifera oil alleviating fat accumulation in high-fat Caenorhabditis elegans,” for consideration in International Journal of Molecular Sciences (MDPI).

OVERALL IMPRESSION

In this study, Liu et al perform an extensive set of transcriptome and metabolome analyses to uncover the potential mechanisms by which CO might alter fat accumulation.  C. elegans is a suitable model for studying alterations in gene expression and metabolism, and the bulk of the paper detailing these experiments is methodologically sound.  However, they fail to clearly establish (or adequately describe) the premise on which these experiments are based – that CO significantly reduces fat accumulation in C. elegans.  I am also concerned with some of the statistical analyses/reporting and an overemphasis on this study’s findings to humans.  Overall, the manuscript has great potential, but I would not recommend it for publication in IJMS until these issues are adequately resolved or clarified.

Major and minor concerns are detailed below:

MAJOR CONCERNS:

1.     Section 2.1 (i.e., Figure 1) is key to this paper since it seeks to establish the fat-reducing effects of CO treatment in high-fat nematodes.  However, it is not convincing in its current form.

a.     There are no statistical analyses reported for Figure 1.  Most staining techniques result in variable staining even WITHIN groups.  So how can we know (with no statistics) if the differences BETWEEN groups are most likely due to chance, or not?  These numbers are even reported in the abstract so this is especially important.

b.     Despite a lack of statistical analysis, Section 2.1 includes the phrases “significantly darker” and “significantly reduced” when describing the differences between groups.

c.     Do the error bars in Figure 1 indicate standard error or standard deviation?  Not clear.

d.     The orlistat group in Figure 1 should be labeled as “Glc-Orlistat” since glucose treatment preceded orlistat treatment (lines 84-85, if my understanding is correct).  Also, there is no mention of the rationale for using orlistat until much later in the paper.

e.     The meanings of the labels above the bars in Figure are unclear (e.g., “a”, “b”, “bc”).

2.     Handling and reporting of statistical analyses are unclear throughout the paper.  Biggest concerns:

a.     Section 4.8 states that “data were expressed as means plus/minus standard deviation.”  However, I can’t find a single place in the paper where data were reported this way.

b.     The only reporting of p values in the paper are in some of the supplementary data tables (S2-S4, S6).  The main figures lack any of this information except for Figure 4C-D where it just indicates p<0.05 in general.

3.     These findings, if built upon, have potential relevance to human health.  However, in its current state, any of these connections should be stated with extreme caution, if at all.  For example, the last sentence of the abstract (lines 26-27) claim that this study provides “valuable insights into the selection and adjustment of dietary oils and highlight the potential health benefits of CO.”  Similar-type claims are made in the last paragraph of the introduction (lines 69-74), the second-to-last paragraph of the discussion (lines 251-252), and the first paragraph of the conclusion (lines 378-381).

   MINOR CONCERNS:

1.     A few descriptions of methodology are insufficient.

a.     Section 4.3 (line 298): “Image J software was applied to density quanitification analyze.”  This statement is unclear and does not adequately describe how this key data set (Figure 1) was analyzed.

b.     Section 4.5 (lines 321-322): The GO and KEGG tools were merely mentioned, with no methodology described.  At minimum, links to these tools should be provided.

c.     Several of the figure captions aren’t descriptive enough and lack basic information needed for independent interpretation.  Figures 1 and 7 are the most significant examples.

2.     Figures 5 and 8 are NOT cited in the body of the text.  The experiments are described, just not cited.

3.     Reference #17 is incomplete.

4.     There are a variety of significant grammatical issues throughout the paper.  Overall, the manuscript is understandable, but the issues are frequent and distracting at times.  It needs proofreading by a native English speaker.

See comments and suggestions, Minor Concern #4.

Author Response

Dear reviewer:

On behalf of all coauthors, I would like to express my sincerely appreciation for giving us a chance to revise our manuscript. Also, I would like to thank all reviewers for their kind and very constructive feedback about our paper. As we regularly do, our resounds to the questions raised were answered point to point. The corresponding revisions were labeled using highlight background. And a clear version was provided.

Response to Reviewer 1 Comments:

MAJOR CONCERNS:

  1. Section 2.1 (i.e., Figure 1) is key to this paper since it seeks to establish the fat-reducing effects of CO treatment in high-fat nematodes. However, it is not convincing in its current form.

  1. There are no statistical analyses reported for Figure 1. Most staining techniques result in variable staining even WITHIN groups.  So how can we know (with no statistics) if the differences BETWEEN groups are most likely due to chance, or not?  These numbers are even reported in the abstract so this is especially important.

Response: Thanks for your suggestion. We have replaced the picture with a scatterplot, which is can be displayed more clearly and accurately in Figure 1 in the revised version.

  1. Despite a lack of statistical analysis, Section 2.1 includes the phrases “significantly darker” and “significantly reduced” when describing the differences between groups.

Response: Thank you for the suggestion. We have performed a statistical analysis in Line 94-98 in the revised version.

  1. Do the error bars in Figure 1 indicate standard error or standard deviation? Not clear.

Response: Thank you for the suggestion. The error bars indicate SD in Line 94-98 in the revised version.

  1. The orlistat group in Figure 1 should be labeled as “Glc-Orlistat” since glucose treatment preceded orlistat treatment (lines 84-85, if my understanding is correct). Also, there is no mention of the rationale for using orlistat until much later in the paper.

Response: According to your advice, Orlistat has been changed to Glc-Orlistat in Figure 1. The rationale for using orlistat is demonstrated in the results in Line 85-87 in the revised version.

  1. The meanings of the labels above the bars in Figure are unclear (e.g., “a”, “b”, “bc”).

Response: Thank you for the suggestion. Letters (a, b) indicate significant differences. We have corrected the description of letters in Figure 1 and Line 94-98 in the revised version.

  1. Handling and reporting of statistical analyses are unclear throughout the paper. Biggest concerns:

  1. Section 4.8 states that “data were expressed as means plus/minus standard deviation.” However, I can’t find a single place in the paper where data were reported this way.

Response: The data of ORO staining intensity and TG content were expressed as means ± standard deviation (SD). We have corrected the description in Line 383-387 in the revised version.

  1. The only reporting of p values in the paper are in some of the supplementary data tables (S2-S4, S6). The main figures lack any of this information except for Figure 4C-D where it just indicates p<0.05 in general.

Response: Thank you for your suggestion, this is our negligence, we have added the corresponding information in the legend of Figure 4, Figure 8 and Figure 9. Line131-135, Line199-200 and Line223-231 in the revised version.

  1. These findings, if built upon, have potential relevance to human health. However, in its current state, any of these connections should be stated with extreme caution, if at all.  For example, the last sentence of the abstract (lines 26-27) claim that this study provides “valuable insights into the selection and adjustment of dietary oils and highlight the potential health benefits of CO.”  Similar-type claims are made in the last paragraph of the introduction (lines 69-74), the second-to-last paragraph of the discussion (lines 251-252), and the first paragraph of the conclusion (lines 378-381).

Response: Thank you for the suggestion. We have changed the corresponding statement to be more cautious. Line26-27, Line71-74 and Line396-397 in the revised version.

   MINOR CONCERNS:

  1. A few descriptions of methodology are insufficient.

  1. Section 4.3 (line 298): “Image J software was applied to density quanitification analyze.” This statement is unclear and does not adequately describe how this key data set (Figure 1) was analyzed.

Response: We have made the necessary modifications in Figure 1 and Line 94-98 in the revised version.

  1. Section 4.5 (lines 321-322): The GO and KEGG tools were merely mentioned, with no methodology described. At minimum, links to these tools should be provided.

Response: We have provided links to GO and KEGG following your advice in Line 388-340 in the revised version.

  1. Several of the figure captions aren’t descriptive enough and lack basic information needed for independent interpretation. Figures 1 and 7 are the most significant examples.

Response: We have added clearer figure captions to the corresponding pictures according to your advice in Line 95-98, Line 133-135, Line 195-197, Line 198-200, Line 223-231 in the revised version.

  1. Figures 5 and 8 are NOT cited in the body of the text. The experiments are described, just not cited.

Response: This is our negligence, and Figures 5 and 8 have been cited in the corresponding content of the manuscript. Line 148 and Line 203-205.

  1. Reference #17 is incomplete.

Response: Reference 17 has been completed. Line 453

  1. There are a variety of significant grammatical issues throughout the paper. Overall, the manuscript is understandable, but the issues are frequent and distracting at times.  It needs proofreading by a native English speaker.

Response: Thank you for the suggestion. The English writing was improved by a postdoc with native English language. the grammatical errors and confused description have also corrected. All the changes were highlighted in the revised version.

Reviewer 2 Report

The authors present a transcriptomics and metabolic analysis of fat accumulation in high-fat C. elegans (made so by glucose treatment) animals as a function of exposure to Camellia oil (CO). The main observations are as follows:

Treatment with CO or orlistat (a lipase inhibitor) reduces fat accumulation in glucose (Glc) treated C. elegans as shown by oil red O staining and measurement of triglycerides (TG) using a commercially available kit.

RNAseq was performed on control, Glc and Glc-CO animals. The authors define DEGs and show Venn diagrams and volcano plots. A GO enrichment analysis is presented identifying various biological processes represented by the DEGs. A subset of DEGs was confirmed by qPCR.

Metabolomics analysis identifies several metabolites up and down in abundance and a PCA is presented on these. Some 114 differentially expressed metabolites were identified. A cluster analysis is performed between the Glc and Glc-CO animals and a KEGG enrichment analysis is presented. Correlative analysis of the DEGs and DEMs is presented. Various pathways are found, for example in purine metabolism and amino acid catabolism and biosynthesis.

The Discussion summarizes the work performed and some of the conclusions reached by the DEG and DEM analysis. For example, the authors conclude that downregulation of fat-7 is the mechanism by which CO treatment reduces fat stores.

Overall, the manuscript presents some omics type data generated in Glc-fed C. elegans animals +/- treatment with CO.

Comments:

1. The authors conclude that downregulation of fat-7 is the mechanism by which CO treatment reduces fat stores, however this is more of a hypothesis to be tested. It predicts, for example, that a fat-7 mutant should show no further fat reduction upon CO treatment. The authors should perform this experiment to test this.

2. The authors also conclude that major pathways of amino acid metabolism are affected by CO treatment. Is there anything that could be discussed? For example, are there other examples in the literature of amino acid metabolism being changed because of CO treatment or other perturbations of lipid storage? Are these changes causative of fat reduction of are they secondary results of reductions in availability of chemical energy through fat catabolism?

3. Does CO treatment by itself (i.e. in wild-type worms, not Glc worms) have any detectable effects on fat stores? How about in calorie-restricted animals?

4. It was surprising not to see any mention of the metabolism network work of the Walhout Lab. The authors should investigate possible relationships of their findings to the extensive metabolome analysis done by this lab. Papers include the following:

https://doi.org/10.15252/msb.20209649

https://doi.org/10.15252/msb.202211443

https://www.sciencedirect.com/science/article/pii/S2589004222009609

5. There are very few overall conclusions to the work beyond just the identification of various differences in gene expression and metabolic flux. It would be good if the authors could propose some meaningful conclusion to the work that is relevant to the original question being asked about the mechanism of fat reduction by CO treatment.

Other comments:

Line 60, use ‘It is’ instead of contraction It’s

Line 76, fix formatting to continue text under Results rather than go to the next page

Fig. 1A,B – Y-axis of histogram in A has ‘Relative’ misspelled as ‘Realive’. P-values should be given between control and Glc, Glc and Glc-CO, etc. and indicated either in the figure or figure legend. A better way to present these data is to use a vertical scatter plot rather than a histogram. This figure could also be made smaller and the images of the worms made larger.

Figure 2 – the correlation coefficient figure does not need to be this large.

Line 151 – space needed in ‘C.elegans’

Page 9, formatting should be done so that large gaps of unused space do not occur.

Figure 9, the top part contains unreadable text. Is the figure necessary or could it be put in supplemental data?

Line 222, description of orlistat should be given in the results when it is first introduced, rather than in the discussion.

Line 277, C. elegans should be italicized here

Line 298-299, the exact number of animals and the numeric SEM values should be reported for each of the bars in the histograms.

Line 300ff / section 4.4, the TG assay should be better described. What is the name of the kit or its inventory number? Is there a reference for how the kit works?

I recommend a native English speaker go over the writing. For example, lines 90-91 the meaning is not clear and the grammar is not correct; line 131 is a sentence fragment “KEGG pathway enrichment analysis of DEGs showed that the biological function.”; line 235, ‘fat-7 mutant’ should be ‘fat-7 mutants’

Author Response

Dear reviewer:

On behalf of all coauthors, I would like to express my sincerely appreciation for giving us a chance to revise our manuscript. Also, I would like to thank all reviewers for their kind and very constructive feedback about our paper. As we regularly do, our resounds to the questions raised were answered point to point. The corresponding revisions were labeled using highlight background. And a clear version was provided.

Response to Reviewer 2 Comments:

  1. The authors conclude that downregulation of fat-7 is the mechanism by which CO treatment reduces fat stores, however this is more of a hypothesis to be tested. It predicts, for example, that a fat-7 mutant should show no further fat reduction upon CO treatment. The authors should perform this experiment to test this.

Response: Thank you for the suggestion. Because we don't have fat-7 mutant strains for the time being due to some factors, this experiment has not been carried out for the time being. This experiment will be considered for further verification in the future.

  1. The authors also conclude that major pathways of amino acid metabolism are affected by CO treatment. Is there anything that could be discussed? For example, are there other examples in the literature of amino acid metabolism being changed because of CO treatment or other perturbations of lipid storage? Are these changes causative of fat reduction of are they secondary results of reductions in availability of chemical energy through fat catabolism?

Response: Thanks for your concern. our research shows that excess glucose significantly affects amino acid metabolism in C. elegans. Therefore, excess glucose help diminish the amino acid load and increases lipogenesis possibly by affecting the TCA cycle. in Line 274-275 in the revised version.

  1. Does CO treatment by itself (i.e. in wild-type worms, not Glc worms) have any detectable effects on fat stores? How about in calorie-restricted animals?

Response: According to previous studies in our lab, CO treatment with a certain concentration had no significant effect on the fat stores of wild-type C. elegans. The effect of CO treatment on calorie-restricted animal has not been studied, but is a good suggestion and will be considered for future studies. Thank you for the suggestion.

  1. It was surprising not to see any mention of the metabolism network work of the Walhout Lab. The authors should investigate possible relationships of their findings to the extensive metabolome analysis done by this lab. Papers include the following:

https://doi.org/10.15252/msb.20209649

https://doi.org/10.15252/msb.202211443

https://www.sciencedirect.com/science/article/pii/S2589004222009609

Response: Thank you for the suggestion. Walhout Lab has done a lot of great and innovative work, which has great reference value for our research and has been discussed in the work in Line 61, Reference 16 and Line 275, Reference 34 in the revised version.

  1. There are very few overall conclusions to the work beyond just the identification of various differences in gene expression and metabolic flux. It would be good if the authors could propose some meaningful conclusion to the work that is relevant to the original question being asked about the mechanism of fat reduction by CO treatment.

Response: Integrated analysis of transcriptome and metabolome the most significantly enriched pathway of Glc-CO/Glc in Figure 9E and Line 218-222 in the revised version.

Other comments:

Line 60, use ‘It is’ instead of contraction It’s

Response: According to your advice, It’s has been changed to ‘It is’, In the revised in Line 60 in the revised version.

Line 76, fix formatting to continue text under Results rather than go to the next page

Response: We have made the appropriate changes according to your advice. Line 75 in the revised version.

Fig. 1A,B – Y-axis of histogram in A has ‘Relative’ misspelled as ‘Realive’. P-values should be given between control and Glc, Glc and Glc-CO, etc. and indicated either in the figure or figure legend. A better way to present these data is to use a vertical scatter plot rather than a histogram. This figure could also be made smaller and the images of the worms made larger.

Response: We have made the appropriate changes and change Figure 1 to a scatter plot according to your advice. In Figure 1 in the revised version.

Figure 2 – the correlation coefficient figure does not need to be this large.

Response: Thank you for the suggestion. We have made the appropriate changes according to your advice in Figure 2 in the revised version.

Line 151 – space needed in ‘C.elegans’

Response: Thank you for the suggestion. We have made the appropriate changes according to your advice in Line 158 in the revised version.

Page 9, formatting should be done so that large gaps of unused space do not occur.

Response: Thank you for the suggestion. We have made the appropriate changes according to your advice. Page 8 in the revised version.

Figure 9, the top part contains unreadable text. Is the figure necessary or could it be put in supplemental data?

Response: Thank you for the suggestion. We have supplemented the corresponding content in the legend. Line 224-231 in the revised version.

Line 222, description of orlistat should be given in the results when it is first introduced, rather than in the discussion.

Response: Thank you for the suggestion. We have implemented the necessary modifications based on your advice in Line 85-87 in the revised version.

Line 277, C. elegans should be italicized here

Response: Thank you for the suggestion. We have made the appropriate changes in Line 85-87 in the revised version.

Line 298-299, the exact number of animals and the numeric SEM values should be reported for each of the bars in the histograms.

Response: Thank you for the suggestion. We have implemented the necessary modifications based on your advice in Line 294-295 in the revised version.

Line 300ff / section 4.4, the TG assay should be better described. What is the name of the kit or its inventory number? Is there a reference for how the kit works?

Response: The name and inventory number of the kits have been added according to your advice. Line 318-320 in the revised version.

Round 2

Reviewer 1 Report

Review of revised manuscript for Liu et al (2023), “Integrated analysis of transcriptome and metabolome provides insight into the Camellia oleifera oil alleviating fat accumulation in high-fat Caenorhabditis elegans,” for consideration in International Journal of Molecular Sciences (MDPI).

RESULTS OF MAJOR REVISIONS

After revision, the manuscript is greatly improved.

1.     Connections to human health, etc. are more cautiously stated.

2.     Better descriptions of some of the results and statistics allow for a clearer and more confident interpretation.  Most critically, Figure 1A is clearer and more convincing.  However, the reported amount of fat reduction found via ORO intensity and TG content (Figure 1) were slightly different in the revised version (compare lines 18 & 85-87 in version 1 with lines 18 & 81-84 in version 2).  I assume the interpretation would be the same, but it could be helpful if the authors provided an explanation for this (not in the paper, but editorially).

3.     The English has greatly improved with only minor issues remaining.

I appreciate the changes the authors made in the revised manuscript and would now recommend acceptance of the manuscript.

Respectfully, I will offer non-binding recommendations for a few noticeable issues:

1.     Line 60: duo due

2.     Line 73: and

3.     Line 97: post-hoctest post-hoc test

4.     Line 170: (Figure S1). indicating (Figure S1), indicating

5.     While, the DEMs of tretinoin significantly downregulated The DEMs of tretinoin were also significantly downregulated.

6.     Line 251: desaturases, has the function of promoting fat stores. desaturases, which function to promote fat stores.

7.     Line 306: according to as

8.     Line 315: This assay at least At least

9.     Line 384: Data of ORO

Author Response

Dear reviewer:

On behalf of all coauthors, I would like to express my sincerely appreciation for giving us a chance to revise our manuscript. Also, I would like to thank all reviewers for their kind and very constructive feedback about our paper. As we regularly do, our resounds to the questions raised were answered point to point. The corresponding revisions were labeled using highlight background. And a clear version was provided.

  1. Better descriptions of some of the results and statistics allow for a clearer and more confident interpretation. Most critically, Figure 1A is clearer and more convincing.  However, the reported amount of fat reduction found via ORO intensity and TG content (Figure 1) were slightly different in the revised version (compare lines 18 & 85-87 in version 1 with lines 18 & 81-84 in version 2). I assume the interpretation would be the same, but it could be helpful if the authors provided an explanation for this (not in the paper, but editorially).

Response: Thanks for your suggestion. The reported amounts of fat reduction found via Oil Red O (ORO) intensity were slightly different in the revised version (compare lines 18 & 85-87 in version 1 with lines 18 & 81-84 in version 2). This difference arises because version 1 utilizes the mean ± standard deviation of three replicates, while version 2 uses solely the mean ± standard deviation of Figure 1A with 30 worms.

  1. Line 60: duo due
  2. Line 73: and
  3. Line 97: post-hoctest post-hoc test
  4. Line 170: (Figure S1). indicating (Figure S1), indicating
  5. While, the DEMs of tretinoin significantly downregulated The DEMs of tretinoin were also significantly downregulated.
  6. Line 251: desaturases, has the function of promoting fat stores. desaturases, which function to promote fat stores.
  7. Line 306: according to as
  8. Line 315: This assay at least At least
  9. Line 384: Data of ORO

Response: Thank you for the suggestion. We have made the appropriate changes according to your advice in Line 60, Line72-74, Line97, Line170, Line 221, Line 221, Line 251, Line 306 Line 315 and Line 384 in the revised version.

Reviewer 2 Report

The authors have satisfactorily addressed my comments.

Author Response

Dear reviewer:

On behalf of all the coauthors, I would like to express our sincere appreciation for diligently reviewing our manuscript and providing numerous precise suggestions, which significantly enhanced its quality.